# High-Throughput Gel Microbeads as Incubators for Bacterial Competition Study

**DOI:** 10.3390/mi14030645

**Published:** 2023-03-12

**Authors:** Trang Anh Nguyen-Le, Xinne Zhao, Michael Bachmann, Philip Ruelens, J. Arjan G. M. de Visser, Larysa Baraban

**Affiliations:** 1Institute of Radiopharmaceutical Cancer Research, Helmholtz-Zentrum Dresden-Rossendorf e. V. (HZDR), 01328 Dresden, Germany; 2Tumor Immunology, University Cancer Center (UCC), University Hospital Carl Gustav Carus Dresden, Technische Universität Dresden, 01307 Dresden, Germany; 3National Center for Tumor Diseases (NCT), Faculty of Medicine and University Hospital Carl Gustav Carus, Technische Universität Dresden, 01307 Dresden, Germany; 4German Cancer Research Center (DKFZ), 69120 Heidelberg, Germany; 5German Cancer Consortium (DKTK), 01309 Dresden, Germany; 6Department of Genetics, Wageningen University, Droevendaalsesteeg 1, 6708 PB Wageningen, The Netherlands; 7Center for Advancing Electronics Dresden (cfaed), Technische Universität Dresden, 01069 Dresden, Germany

**Keywords:** millifluidic, high-throughput, agarose microbeads, co-culture, bacterial co-existence, fluorescence-tagged *E. coli*

## Abstract

Bacteria primarily live in structured environments, such as colonies and biofilms, attached to surfaces or growing within soft tissues. They are engaged in local competitive and cooperative interactions impacting our health and well-being, for example, by affecting population-level drug resistance. Our knowledge of bacterial competition and cooperation within soft matrices is incomplete, partly because we lack high-throughput tools to quantitatively study their interactions. Here, we introduce a method to generate a large amount of agarose microbeads that mimic the natural culture conditions experienced by bacteria to co-encapsulate two strains of fluorescence-labeled *Escherichia coli*. Focusing specifically on low bacterial inoculum (1–100 cells/capsule), we demonstrate a study on the formation of colonies of both strains within these 3D scaffolds and follow their growth kinetics and interaction using fluorescence microscopy in highly replicated experiments. We confirmed that the average final colony size is inversely proportional to the inoculum size in this semi-solid environment as a result of limited available resources. Furthermore, the colony shape and fluorescence intensity per colony are distinctly different in monoculture and co-culture. The experimental observations in mono- and co-culture are compared with predictions from a simple growth model. We suggest that our high throughput and small footprint microbead system is an excellent platform for future investigation of competitive and cooperative interactions in bacterial communities under diverse conditions, including antibiotics stress.

## 1. Introduction

A wide variety of microbial species coexist in the natural environment, and their balance is essential to maintaining human well-being [1]. For instance, the complex microbial communities in the oral cavity and gut contribute to our health [2,3], while those communities in soil and food influence our livelihood [4,5]. In nature, bacterial communities constantly interact with surrounding neighbors to share scarce nutrients and limited space. Various model systems have been developed to understand multiple scenarios and mechanisms of microbial interactions in pure and mixed cultures [6,7,8,9]. Most findings agreed that a structured environment, such as biofilms and semi-solids (e.g., food, soil) is one of the most important factors that determines the scope and scale of competitive-cooperative interaction [10,11]. Even though current technology allows the efficient isolation and cultivation of bacteria, species removed from their natural environment may express different behavior in artificial culture [12]. The dissimilarity is partly due to the lack of inter-species interactions and spatial distribution, which directly influence their evolutionary dynamics [13]. Thus, the main challenges are: (a) to mimic essential aspects of natural environments, such as their spatial structure and community composition, and (b) to develop high throughput systems to analyze their population dynamics.

Regarding the first challenge, a solid culture within, e.g., hydrogels, is an optimal option for simulating the spatial confinements of microbes [4,14]. For instance, Johnston et al. used a temperature-responsive and shear-thinning hydrogel to culture bacteria and collect small molecule and peptide production in microbial monoculture and consortia. [15]. Moreover, Ming’s group utilized the hydrogel, which has a 3D molecular network structure and high biocompatibility, and provides a moist environment to culture beneficial bacteria for wound healing [16].

Regarding the second, microfluidic technology fulfills the requirements to create multiple identical reactors for the statistical analysis of microbial co-cultures. A combination of both, high-throughput techniques using hydrogel media, offer significant advantages and have been applied in several studies, such as screening of the expressed genes by fluorescence-activated cell sorting [17,18,19], studying interactions between antibiotics and associated mutants [20], imaging of single-cell [21]. Further, the bioprinting technique is used to form hydrogel spatial segregation between co-culture bacterial strains. For instance, Kumar [22] developed a printing method to arrange bacterial genotypes across a sub-millimeter array of emulsion droplets. Ceballos-González [23] printed spatial controlled bacterial microcosms using alginate and calcium chloride. However, the combination of micro- and milli-fluidic tools and solid media is still not fully explored for bacterial community co-culture systems [18]. Especially, while high-density mixed populations are easily studied in droplet settings to observe collective effects, emergent phenomena of cooperation and competition, as well as final co-existence equilibria, are visible at low inoculum (1–100 cells/container), which is typically investigated in solid media.

In this work, we generate nanoliter-volume micro gel beads that are positioned as an excellent platform to study competitive and cooperative interactions between two strains of the bacterium *Escherichia coli* (*E. coli*), growing as colonies. For this, we use a millifluidic assembly [24] and off-shelf components with the addition of low-melting agarose in the culture medium. Unlike picolitre-volume, microfluidic droplet systems or standard agar plates (volume of ca. picoliters or a few dozens of milliliters, respectively), our agarose microbeads represent an intermediate size of ca. hundreds of nanoliters. Millifluidic-scale culture volumes are required to study population-level consequences of microbial interactions, as detecting changes in population dynamics requires observations across multiple generations in exponentially growing populations [24]. The two *E. coli* strains we use in our study are genetically identical except for a chromosomal gene encoding blue and yellow fluorescence. We track their growth kinetics by monitoring colony size, shape, and fluorescence intensity changes using fluorescence microscopy. The two strains are co-cultured at a one-to-one ratio while their growth kinetics are compared to those in monoculture. Interactions are analyzed by comparing the strains’ colony size, occupancy of the microbead, and intensity of produced fluorescent signal per cell during growth. Finally, we use a theoretical model to predict colony size growth kinetics in the co-culture system based on monoculture growth kinetics.

## 2. Materials and Methods

### 2.1. Materials

*E. coli* strain MG1655 (MG1655 galK::SYFP2-FRT) was genetically modified to possess a chromosomal SYFP2 gene or chromosomal mTagBFP2 gene and to exhibit blue or yellow fluorescence [24,25]. Bacteria stock was prepared as described in Appendix A. We employed an autofluorescence-free M9 minimal salt culture medium in all experiments. All materials were sterilized before use. We ultilized the following reagents and materials: D-glucose monohydrate (1083421000, Millipore, Germany), casein hydrolysate (22090, Sigma Aldrich, Germany), magnesium chloride (2008337, Sigma Aldrich, Germany) ultra-low gelling temperature agarose (A2576, Sigma Aldrich, Germany), hydrofluoether (HFE) oil (Novec 7500, IoLiTec Ionic Liquids Technologies GmbH, Germany), surfactant (PicoSurf 2^TM^ 2% (*w/w*), Dolomite, UK), and mineral oil (M5904, Sigma Aldrich, Germany). The fluidic system was constructed using transparent Fluorinated ethylene propylene (FEP)-tubings, Ethylene tetrafluoroethylene (ETFE) T-junction, and cross-junction, 2-way manual valves (1/16” OD, IDEX Health & Science, USA). The flow rate in the fluidic system was controlled by a 4-dosing units precision pump Nemesys (CETONI, Germany).

### 2.2. Encapsulation of Bacteria

Bacteria concentration was first estimated by optical density measurement (BioPhotometer 6131, Eppendorf, Germany). We generated gel microbeads with encapsulated bacteria by injecting the bacteria solution into the fluidic circuit represented in Figure 1a and Appendix A. We fixed the flow rate for bacteria, agarose, HFE, and mineral oil at 1.5, 1.5, 1, and 7 mL/h, respectively, while maintaining the agarose solution temperature at 60 °C. After generation, ca. 200 droplets were stored in the reserving coil (Appendix A). The reserving coil was then removed and cooled down at 4 °C for 15 min to induce gelation. The gel beads were then flushed out from the coil and placed on a sterile 35 mm glass petri dish for incubation and imaging, covered by a thin layer of mineral oil. The mineral oil was essential to prevent evaporation and shrinkage of the gel beads. We placed the petri dish in a mini-incubator (VWR) for bacteria culture at 37 °C. The complete experimental setup is shown in Appendix A.

### 2.3. Image Acquisition and Analysis

Images of the gel beads were taken under an inverted fluorescent microscope (Carl Zeiss Axiovert 200 M, Germany) using two sets of filters corresponding to the excitation and emission spectrum of *E. coli* YFP and *E. coli* BFP. Brightfield, yellow fluorescence, and blue fluorescence images were recorded at multiple locations of the sample. All images were processed with ImageJ software. A brief description of the image analysis was shown in Appendix A. Data from these images are further analyzed using MATLAB (version 2019a, Germany) and Origin Pro (version 9.0, Germany). The modeling and fitting of bacteria growth curve was performed on Python (3.8., USA)

### 2.4. Determination of Bacteria Colony Size and Modelling

To estimate the bacteria colony size in agarose beads and its changes, the colony size (experimental data points of colony obtained under the fluorescence microscope) of two strains of *E. coli* monoculture and co-culture was first fitted to a sigmoidal curve based on the Boltzmann function Equation (1) and shown in Appendix A. In all the models, colony size *A*(*T*) is assumed to be proportional to the cell number *N* (*A* ∝ *N*).
(1)$$A\left(T\right)=A_{2}+\left(A_{1}-A_{2}\right){\left(1+e^{\frac{T-T_{0}}{dT}}\right)}^{-1}$$
where *A = A(T)* is the colony size (area) at time point *T*; *T*_0_ represents the initial time point; *A*_2_ is the final colony size, and *A*_1_ is the initial colony size. Fitting of monocultures and co-cultures was performed from 4 h of incubation (when colonies were observed) to the stationary phase and used for subsequent data modeling to reduce error amplification caused by discontinuities in the experimental data. Data modeling was predicted with Ram’s model of describing the bacterial growth in co-culture [26]. The monoculture model (Baranyi–Robert model [26], see Equation (2)) describes the bacterial growth rate change as cells adjust to new conditions and resources become scarce or depleted, which can also be displayed in the form of Equation (3). This monoculture model was first separately fitted to monoculture colony size fit curves of two bacterial strains, obtaining the parameters of monoculture in Equations (2) and (3) [27,28,29].
(2)$$\frac{dA}{dT}=rA\left(1-{\left(\frac{A}{A_{2}}\right)}^{\gamma }\right)$$
(3)$$A\left(T\right)=\frac{A_{2}}{{\left(1-\left(1-{\left(\frac{A_{2}}{A_{1}}\right)}^{r}\right)e^{-r\gamma T}\right)}^{1/\gamma }}$$
where *A* = *A(T)* is the colony size at time point *T*, *A*_1_ is the initial colony size, *A*_2_ is the maximum colony size, *r* is the initial per capita growth rate, and *γ* is a deceleration parameter, which describes the transition rate from fast growth to slow growth.

Then the co-culture model (2-strain Lotka–Volterra competition models [27], see Equations (4)–(6)) was fitted to the summation of co-culture colony size fitted curves. The co-culture predicting colony size curves are displayed in Appendix A to compare experimental data and its sigmoid fitting curves.
(4)$$\frac{dA_{B}}{dT}=r_{B}A_{B}\left(1-\frac{A_{B}^{\gamma_{B}}}{A_{2,\;B}^{\gamma_{B}}}-c_{Y}\times \frac{A_{Y}^{\gamma_{Y}}}{A_{2,\;B}^{\gamma_{B}}}\right)$$
(5)$$\frac{dA_{Y}}{dT}=r_{Y}A_{Y}\left(1-c_{B}\times \frac{A_{B}^{\gamma_{B}}}{A_{2,\;Y}^{\gamma_{Y}}}-\frac{A_{Y}^{\gamma_{Y}}}{A_{2,\;Y}^{\gamma_{Y}}}\right)$$
(6)$$A_{total}=A_{B}+A_{Y}$$
where *B* represents BFP, while *Y* represents the YFP strain, *A_B_* is the colony size of *E. coli* BFP and *A_Y_* is the colony size of *E. coli* YFP. Since the parameters, *r_B_*, *r_Y_*, *A*_1,*B*_, *A_2,B_*, *A*_1,*Y*_, *A_2,Y_*, *γ_A_*, and *γ_B_* can be obtained from the monoculture modeling results and *A_total_* is the summation of the colony sizes of the two strains (co-cultured), the individual growth curves of the two co-cultured *E. coli* strains can be simulated.

## 3. Results and Discussion

### 3.1. Agarose Microbeads for Bacteria Encapsulation

In this work, we investigated the colony growth of co-cultured bacteria by embedding two strains of *E. coli*, constitutively expressing yellow fluorescent proteins (YFP) and blue fluorescence (BFP) [24] into hundreds of agarose microbeads. The gelation process of agarose beads was thermally activated. Low temperature melting agarose (1.5% *w/w*) with a melting point at 60 °C and gelation at ca. 20 °C was added to the nutrient culture medium M9. For the following experiments, we obtained the monodispersed gel beads of 240 ± 10 nL that correspond to 1.2 ± 0.05 mm linear dimensions (Appendix A). The setup for the formation of microbeads is depicted in Figure 1a. The three immiscible fluids adjoined at the cross-junction and form a sequence of aqueous agarose droplets and mineral oil interspaces carried by HFE oil (Appendix A). The aqueous droplets were reserved in an FEP tubing coil and later cooled down to initiate the solidifying process of the agarose. In the current setup, the system produces approximately 200 mini-agarose beads per minute (Appendix A) with minimum effort and consumables. Such a configuration is suitable for investigating microbial population dynamics in semi-solid environments in a high-throughput manner and offers high flexibility in tuning the study parameters, e.g., variation of inoculum size.

While bacteria in liquid droplets freely grew in the planktonic form, *E. coli* inside of the structured agarose microbeads formed tight three-dimensional colonies (Figure 1b). Since the strains of *E. coli* express YFP and BFP, Figure 1 clearly shows a fluorescence corresponding to their colonies’ appearance 12 h after inoculation. Detailed analysis of colony morphology is provided in a later section.

Considering that each colony usually started from a single bacterium [4], we could infer the inoculum size and distribution by observing the colonies across the gel beads (Figure 1c,d). As encapsulation at low inoculum size is our priority in this work, the potential shading, i.e., colonies growing behind other colonies in 3D, has a negligible effect. The two strains were mixed with a ratio of 1:1 and encapsulated in microbeads at a final concentration of two cells per droplet, aiming for one *E. coli* YFP and one *E. coli* BFP per bead. The experimental results show a realized average encapsulation of 2.56 cells/microbead, slightly higher than the expected inoculum of two cells/microbead, calculated assuming a Poisson distribution (Figure 1c). This discrepancy is due to the deviation of the estimated volume (200 nL) in the liquid droplet [24] and the actual experimentally confirmed data (240 nL) in gel beads at the same flow rate (Appendix A).

Finally, we investigated the distribution of bacterial communities across the bead. The heatmap in Figure 1d reveals that the frequencies of beads containing two strains (co-culture), one strain (monoculture), and zero bacteria are 55.8%, 34.9%, and 9.3%, respectively. Although the number of beads consisting of precisely one *E. coli* YFP and one *E. coli* BFP was comparable at inocula of 1.35 (13.7%) and 2.56 cells/bead (14.1%), the co-culture condition at an inoculum of 1.35 cells/bead was only 26.1%, almost half of that of 2.56 cells/bead (Appendix A). These experimental results agree with the Monte Carlo simulation result, which indicates a clear transition from monoculture to co-culture dominance at inocula higher than two cells/bead (Appendix A). In the following, since our analysis is performed on the whole bead population, we focused on working at an inoculum of 2.56 cells/bead to maximize the ability to observe the effect of bacterial co-existence.

### 3.2. Colony Morphology in Agarose Microbeads

Unlike a liquid medium, a stiff agarose matrix does not support the detachment of the cells during division and their diffusional walk after division, which forces the bacteria to grow into colonies [30]. It is expected that the change of the stiffness in the hydrogel matrix will dramatically influence the size and the shape of the resulting colonies, causing the smooth transition from the equally distributed microbes in the aqueous medium (e.g., 0–0.5% of agarose content), to a tight assembly of the microbes at, e.g., 1.5%. This is reflected in Figure 1e, demonstrating the evolution in the distribution of the microbes within the beads upon an increase in the agarose concentration. Obviously, a stiffer matrix of 1.5% agarose, gel strength ~400 g/cm^2^) can initiate the formation of the tightly packed colony compared to the delocalized assemblies observed in the beads with the 0.5 and 1% (gel strength < 100 g/cm^2^) of the gel [31]. Although our system can reliably produce gel beads with the mono-dispersed size at agarose concentration up to 3% (Appendix A), further experiments focus on the case of the 1.5% agarose as an optimal configuration to achieve the finite size colonies and the reported optimal environment to mimic the in vivo conditions.

Growth of the colonies was monitored and analyzed using brightfield and fluorescence microscopy (Figure 2a). Surprisingly, both strains of *E. coli* formed asymmetric colonies inside microbeads instead of a round shape like most observations on the surface of the agar medium [30]. The analysis is conducted on colonies that grew for 24 h. Two dimensionless quantities describing the shape of the colony are calculated as follows:(7)$$C=4\pi \times \frac{\left[\mathrm{area}\right]}{{\left[\mathrm{perimeter}\right]}^{2}}$$
(8)$$AR=\frac{\left[\mathrm{major axis}\right]}{\left[\mathrm{minor axis}\right]}=\frac{\left[\mathrm{area}\right]}{\pi \times {\left[\mathrm{minor axis}\right]}^{2}}$$

Figure 2a presents examples of *E. coli* colonies observed in the microbeads with the corresponding value of the descriptors. Circularity (*C*) with a value of 1.0 indicates a perfectly circular shape, while values close to 0.0 indicate an increasingly elongated polygon. Meanwhile, the aspect ratio (*AR*) gives a sense of symmetry to the colony by comparing the longitudinal and transverse dimensions of the shape. In addition, Figure 2c,d show histograms of the frequency of each shape descriptor over 423 *E. coli* BFP and 371 *E. coli* YFP colonies.

In summary, the colony shape of the two bacterial strains in monoculture is almost identical. The shape descriptors show that the *E. coli* grew into an ellipsoidal shape with an aspect ratio of around 1.3. The elongated *E. coli* colony was also observed in a confined environment elsewhere [32,33]. We propose that this asymmetric pattern is associated with the rod-like structure of *E. coli* and the directionality of the process of cell division. The transition of the *E. coli* colony to a more isotropic shape is nicely explained in [32]. Briefly, the colony would expand longitudinally along a common axis until bothered by defects and the cells push outwards in all directions. Therefore, the ellipsoidal colony is more likely in a confined semi-solid gel system, which is supported by studies in a macroscopic petri dish. For the latter, we compared the shapes of bacterial colonies grown at the surface of the agar (round-shaped) and colonies grown from bacteria suspended within the agar (elliptical shape) (see Appendix A).

Note that the conventional microscopic images can only provide a 2D representation of the colonies, while *E. coli* form a 3D structure. Indeed, both strains formed ellipsoidal colonies in microbeads, which were confirmed with confocal microscopy (Figure 2b, Appendix A). Assuming a colony as a perfect ellipsoid, the volume of the colony can then be calculated by the equation:(9)$$V_{colony}=\frac{4}{3}\times \pi \times \frac{\left[\mathrm{major axis}\right]}{2}\times {\left(\frac{\left[\mathrm{minor axis}\right]}{2}\right)}^{2}$$

Combining (Equations (8) and (9)), we have:(10)$$V_{colony}=\frac{\pi }{6}\times AR\times {\left(\frac{4}{\pi }\times \frac{\left[\mathrm{area}\right]}{AR}\right)}^{3/2}$$

Interestingly, the colony morphology of two *E. coli* strains developed differently in different culturing conditions inside agarose beads (Figure 2e–h and Appendix A). *E. coli* YFP colonies grew into a more isotropic shape in co-culture than the BFP colonies (Figure 2g,h). However, despite changes in *E. coli* YFP, the colony shape of *E. coli* BFP colonies in co-culture did not change dramatically (Figure 2e,f). These differences in the colony morphology of near-isogenic *E. coli* strains in co-culture conditions may indicate the influence of the competitive environment under confinement on slight differences in growth rate or cell morphology but need to be further investigated.

### 3.3. Effect of Inoculum on Colony Size

In this section, we describe the impact of initial *E. coli* inoculum size on the final population size and the size of individual colonies. For these experiments, *E. coli* YFP and *E. coli* BFP were mixed 1:1 and were then encapsulated in agarose microbeads and incubated for 24 h with the inoculum size spanning from 1, 2, 10, 100, to 500 cells/bead (Figure 3a). No further growth of colonies was observed after 24 h at any inoculum size.

Figure 3b presents the relationship between the estimated volume of the individual colonies and inoculum size (more details in Appendix A). The data show a clear inverse relationship between the colony size and inoculum, as expected from the constant amount of nutrients per gel bead. Previous reports showed that the pore size of a 1.5% agarose network ranged from 200 to 400 nm [34,35], which is sufficient for the diffusion of nutrients and signaling molecules yet prohibits cell mobility [30]. In this study, glucose is the limiting nutrient source and has a diffusion rate in agarose of approximately 600 μm^2^/s at 25 °C [36,37], only 5% slower than in water [36]. Therefore, we assume that all colonies share equitable access to the growth-limiting nutrient in the microbead. As the same glucose concentration is prepared for all inoculum sizes, the higher the inoculum, the lower the total amount of nutrients received per colony, leading to a proportionally smaller average colony size.

Similarly, Figure 3c illustrates a decline in the estimated number of generations (G) produced in single colonies with increasing inoculum size. The number G is calculated based on several assumptions: (1) a colony developed from a single bacterium via exponential growth; (2) daughter bacteria are densely packed within the colony; (3) a colony forms an ellipsoidal shape whose volume is calculated based on (Equation (10)); and (4) the volume of a *E. coli* BFP and *E. coli* YFP cell is similar and equal to a typical *E. coli* of $V_{E.coli}\approx 1.3\;{\mu m}^{3}$ [38].
(11)$$G={log}_{2}\left(\frac{V_{colony}}{V_{E.coli}}\right)$$

It shows that the current setup can accommodate up to nineteen generations from a single bacterium to approximately 10^6^ cells in each microbead (Figure 3c,d). The numbers correspond to those in liquid droplets with similar glucose concentration [24]. Later, the volumetric occupancy of each *E. coli* strain (O) in the microbead is determined as:(12)$$O=\frac{n}{2}\times \frac{V_{colony}}{V_{droplet}}\times 100\%$$
where $V_{colony}$ is the colony volume, $\frac{n}{2}$ is the inoculum of each strain at a ratio of 1:1, and $V_{bead}$ Is the microbead volume. In addition, an approximate conversion of the total colony volume to total population size in a microbead (N) is calculated as:(13)$$N=N_{E.coliBFP}{+N}_{E.coliYFP}=\frac{\overset{-}{n}}{2}\times \left(\frac{V_{colonyBFP}}{V_{E.coli}}+\frac{V_{colonyYFP}}{V_{E.coli}}\right)$$

Figure 3d shows that the bacteria occupy an infinitesimal portion of the microbead volume, less than 0.5% for both strains combined at all inoculum sizes. The results demonstrate that physical space is not the limiting factor for bacterial growth in our study. On the other hand, the final population size in microbeads is comparable and independent of inoculum size, ranging from 10^5^ to 10^6^ cells/bead. This result further supports that nutrient depletion is the primary factor determining the final colony size.

Interestingly, the *E. coli* YFP’s colonies are slightly larger than those of *E. coli* BFP in most cases (Appendix A). In addition, although preparing at the same concentration, which was confirmed by optical density measurement, the final count of *E. coli* YFP was often higher than that of *E. coli* BFP. The phenomenon is consistent throughout this study and previous work [24] and could be related to a faster growth rate of *E. coli* YFP than *E. coli* BFP. Meballos-González et al. demonstrated that recombinant *E. coli* strains from the same original strain may exhibit different specific growth rates due to the expression of distinct proteins and different metabolic loads, e.g., *E. coli* EcRFP grows faster than *E. coli* EcGFP in the same culture environment [23].

### 3.4. Colony Growth Dynamics

Here, we monitor the growth dynamics of *E. coli* YFP and *E. coli* BFP in agarose microbeads at an inoculum of two cells/bead. In monoculture, we inoculated single *E. coli* strain into the microbeads. In the co-culture system, *E. coli* YFP and *E. coli* BFP were mixed at a ratio of 1:1 to study their interaction. Microscopic pictures were taken every 2 h for the first 12 h and then after 10 and 20 h (Figure 4a). The lag phase of the growth curve could not be determined precisely due to the challenge of identifying the colonies in the first two hours of incubation, as many of them were still below the detection limit of our microscope. Therefore, the pictures from this stage were not analyzed but used as a baseline for fluorescence intensity. From 4 h onward, the size and the intensity of each colony increased detectably and were processed using the method described in the Appendix A. The growth curves were constructed using the colonies’ measured area and fluorescent intensity, as shown in Figure 4b,c.

As a general pattern, the growth of bacterial populations goes through three stages: lag phase, exponential phase, and stationary phase [30,39]. A similar model can be applied to all the curves obtained in Figure 4b,c, using the colony size and increase of total colony fluorescent signals as variables. In addition, the bacteria need time to adapt to the growth condition i.e., the lag phase. During the period from t = 4–10 h, colonies were noticed and entered the exponential phase in most cases. The growth was then slowed down from 10 to 12 h, which can be explained by nutrient depletion. The deceleration phase was also observed in another study of *E. coli* growth kinetics in an agar well plate [30].

From 12 h on, a difference between fluorescent intensity and colony size arose. The colony size remained unchanged in most cases during the next two days (Figure 4b), while the fluorescent intensity decreased significantly (Figure 4c). The reduction of fluorescence signal could be explained by photo-bleaching or decreased fluorescence per cell due to nutrient starvation. On the other hand, a stable colony size after 12 h did not guarantee an endless number of live bacteria. It merely reflected the occupied area of bacteria, but not the vital status of the cells. However, by comparing the two signals (colony size and fluorescence intensity), we could speculate that the time the bacterial populations reached the stationary phase was approximately 12 h. We noticed that the colony size of *E. coli* YFP strains was larger than *E. coli* BFP. The result is consistent in both experimental data (Figure 2b and Figure 4b) and fit curves (see Appendix A) to the model (see Appendix A). The predicted growth curves have a more extended lag phase than experimentally measured due to the absence of the experimental data points for the first 4 h. Next, in the exponential phase, the *E. coli* YFP strain’s growth was faster than that of *E. coli* BFP, as shown by both modeling and experiment. However, as the stationary phase was reached, the deviation of modeling from experimental data arose. The model predicts that the final colony size of *E. coli* YFP would be larger and *E. coli* BFP would reach a smaller size than experimental data, yet they are both within the experimental standard error. Finally, a clear difference between monoculture and co-culture was observed in the stationary phase. We saw that both strains had higher fluorescent intensity in monoculture than in co-culture which suggests mutual inhibition interaction in co-culture condition. In addition, a more significant reduction of fluorescent intensity happened in *E. coli* YFP than *E. coli* BFP. The phenomenon is likely due to the photobleaching of the two fluorescent proteins and the lower stability of the YFP at the later stationary phase [4,30].

### 3.5. Interactions between E. coli YFP and E. coli BFP

In this section, we investigated the possible competitive and cooperative interactions between the two strains by comparing their growth behavior in monoculture and co-culture in microbeads and other detection setups. Although the growth rate of *E. coli* YFP and *E. coli* BFP individually in bulk liquid medium were almost identical (Appendix A), results from agarose microbeads and liquid microwell plate show that *E. coli* YFP grew faster than *E. coli* BFP (Appendix A). Note that the bulk liquid medium results are recorded with optical density while the other two methods measure fluorescent intensity. It suggests that the origin of YFP’s faster growth is related to the fluorescent proteins, which is the only difference between the two strains. The behavior might be a consequence of phototoxicity associated with live-cell imaging and fluorescent proteins. Producing fluorescent signals could introduce more growth-inhibiting stress to one strain than the other. For example, E. C. Jensen’s study shows that BFP may raise slight toxic effects [40] and makes the strain less fit than YFP [41]. In addition, in our previous work, we noticed that the pH value decreased for both monocultured strains during incubation and decreased more for *E. coli* YFP. This difference in pH change may be caused by the different rates of metabolite production due to the difference in growth rates of the two strains, which may relate to subtle differences in the metabolic costs involved in fluorophore expression [24]. In conclusion, the exact reasons causing the YFP strain to grow faster than the BFP strain are unclear but could be due to an increased metabolic burden of the mTagBFP2 protein [23].

An increasing doubling time in co-culture (1:1) compared to monoculture in bulk liquid medium (Appendix A) indicates a negative (i.e., competitive) interaction between the two strains, whereas a decreased doubling time indicates a positive (i.e., cooperative) interaction. We tested the interactions between both strains in agarose microbeads (Appendix A) and liquid microwell plates (Appendix A). The results show that the presence of the other strain significantly increased the doubling time of *E. coli* BFP but had a slightly opposite effect on *E. coli* YFP. Still, both strains could not reach the final fluorescent intensity per colony as high as in monoculture conditions. To further illustrate this interaction, Figure 5 presents scatter plots that show a correlation between the two species’ final cell densities in agarose microbeads constructed based on colony size (Figure 5a) and fluorescent intensity (Figure 5b) at 12 h. The *E. coli* YFP and *E. coli* BFP were randomly paired and compared to the median of the monoculture case. Four sections were mapped out corresponding to the nature of the interactions. In monoculture conditions, we had no way to distinguish between interaction types, and we used observed colony sizes as a reference for comparison with the co-culture results. When adding the co-culture data to the map, both intensity and colony size shifted to the “mutual inhibition” corner, indicating that growth inhibition via competitive interactions prevailed.

## 4. Conclusions

Using a millifluidic system, we introduced agarose medium into gel microbeads with only a 10^−5^ fraction of the volume of traditional solid medium, i.e., a Petri dish with agar medium. The setup is straightforward, high-throughput, and well-controlled, which simultaneously facilitates the study of hundreds to thousands of micro-bioreactors.

The bacterial co-existence study was initiated by encapsulating two fluorescently labeled strains of *E. coli* at a ratio of 1:1. When varying the inoculum size while keeping the volume of the gel beads constant, we obtained a clear inverse relationship of colony versus inoculum size due to increased limitation of available resources per colony within a confined volume. The demonstration showed that our methodology is particularly relevant for investigating bacterial growth kinetics at low inocula by observing the difference in colony morphology, size, and fluorescence intensity. Interestingly, we encountered different growth behavior of the two near-isogenic *E. coli* strains since the YFP strain had a larger colony size, faster growth rate, and more isotropic colony morphology in co-culture with the BFP strain than vice versa. The observation was confirmed by both experimental and modeling results. The results suggest that the *E. coli* YFP strain has a slight growth advantage relative to *E. coli* BFP, which is consistent with previous findings in liquid cultures [40]. Furthermore, by comparing results from co-culture and monoculture, we showed that interactions between *E. coli* YFP and BFP colonies in the gel beads were predominantly competitive, leading to mutual growth inhibition.

In conclusion, our work shows that the miniaturized solid bioreactor we present here may be valuable for exploring the nature of interactions within bacterial communities at high resolution under semi-natural spatially structured conditions. The developed platform operates with the large number of the gel nanoliter reactors, which is of great importance for the field of microbiology. On one hand, major interest comes from the fundamental science, dealing with the evolutionary microbiology e.g., for investigation of the appearance and spreading of the antibiotic resistance within and between the bacterial species. On the other hand, there is a high potential to develop more practical applications, e.g., antibiotics susceptibility, and drug screening assays, performed in semi-solid miniaturized environments.

## Figures and Tables

**Figure 1 micromachines-14-00645-f001:**
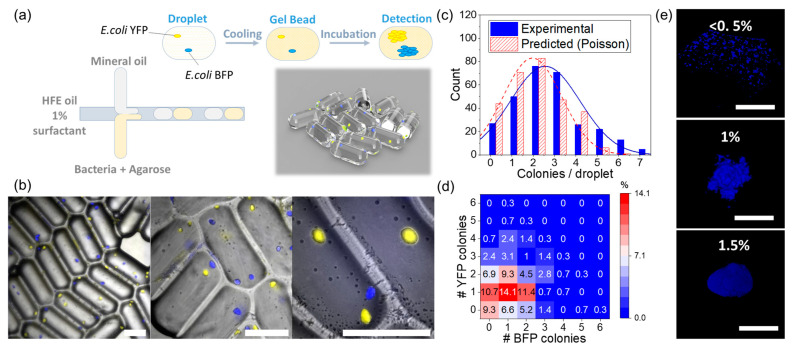
Agarose microbeads as mini-Petri dishes for bacterial co-cultivation study (**a**) Concept illustration of the generation of the agarose microbeads for bacteria co-cultivation. Droplets of liquid agarose mixed with one or two types of bacteria were generated using a millifluidic system. The agarose crosslinks at low temperature and turns droplets into gel beads immobilizing the contained bacteria. (**b**) Overlap images of brightfield and 2-fluorescence-channel of *E. coli* BFP and *E. coli* YFP colonies in gel beads after 12 h of incubation at different magnification (left to right: 2.5×, 5×, 10×). The scale bars represent 500 μm. (**c**) Histogram of bacterial colony number distribution per microbead (droplet) in experimental data and predicted Poisson distribution, showing the number of bacterial colonies per microbead matched the predicated Poisson distribution at an inoculum of 2 cells/microbead. (**d**) Distribution of *E. coli* YFP and *E. coli* BFP colonies in microbeads at an inoculation ratio of 1:1 and an inoculum size of 2 cells/microbead. The color scale from blue to red represents the probability of having a specific colony number of BFP (0–6) or YFP (0–6) cases in the bead, e.g., the statistical probability of having one BFP and one YFP is 14.1%. (**e**) Confocal microscope images of *E. coli* BFP in gel bead at different agarose concentrations (scale bar: 100 μm).

**Figure 2 micromachines-14-00645-f002:**
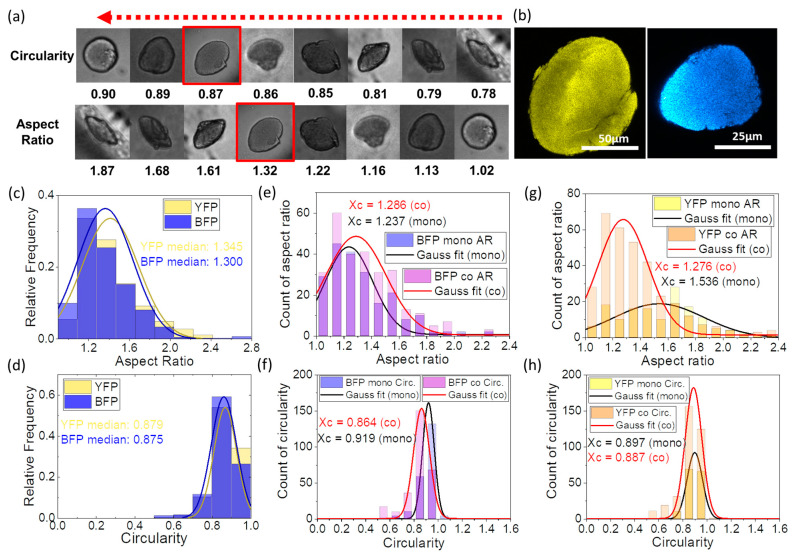
Typical morphology of bacterial colony inside agarose microbeads (**a**) Exemplary images of different bacterial colony morphologies formed in agarose microbeads categorized by shape indicators. Red boxes indicate the type of morphology that presents at the highest frequency after 12 h of incubation in co-culture. (**b**) Confocal microscopy images of *E. coli* YFP and *E. coli* BFP colonies grown in gel beads after 12 h of incubation in co-culture. (**c**) Aspect ratio and (**d**) circularity distribution of *E. coli* YFP and BFP colonies after 12 h of incubation. Comparison of (**e**) aspect ratio and (**f**) circularity of *E. coli* BFP monoculture and co-culture after incubating for more than 20 h. Comparison of (**g**) aspect ratio and (**h**) circularity of *E. coli* YFP monoculture and co-culture after incubating for more than 20 h.

**Figure 3 micromachines-14-00645-f003:**
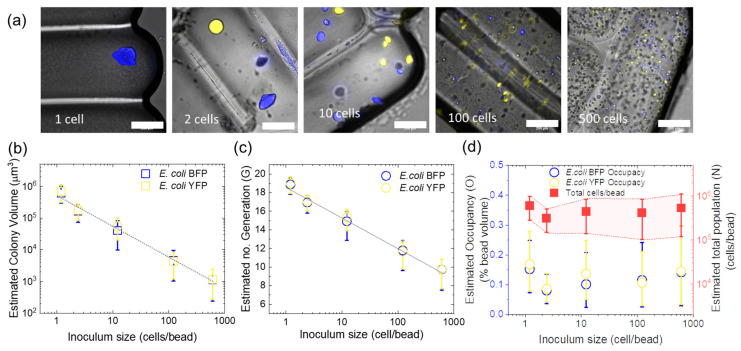
Effect of inoculum size on the final colony size of *E. coli* YFP and *E. coli* BFP in agarose microbeads after 24 h of incubation. (**a**) Overlap microscopic images of bacterial colonies in microbeads at different initial inoculation densities. Scale bar: 200 μm. (**b**) Relationship between estimated bacterial colony volume and inoculum size. (**c**) Relationship between the estimated number of bacterial generations produced in a single colony and inoculum size. (**d**) Volume occupancy of bacterial colonies in the microbead (left axis, yellow and blue circles) and total cell population (right axis, red squares) at different inoculum sizes. Error bars represent errors calculated from the standard deviation of the measured area of at least 59 sample colonies.

**Figure 4 micromachines-14-00645-f004:**
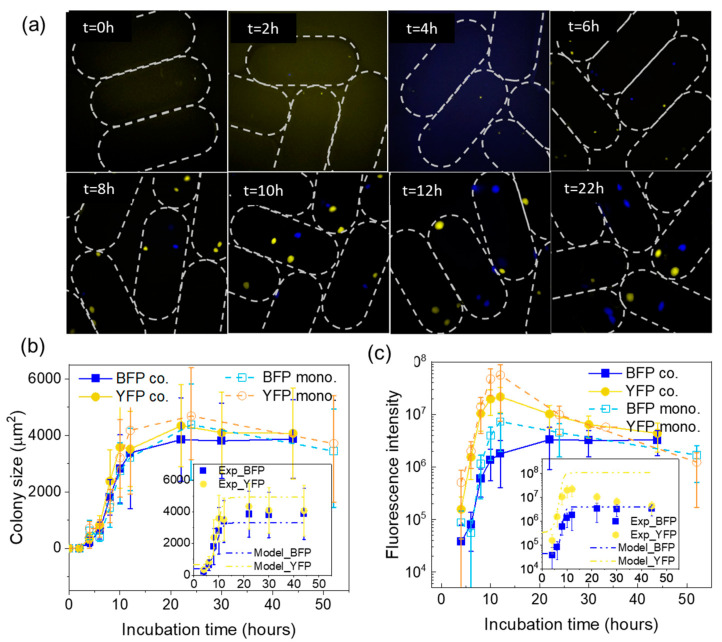
The growth of *E. coli* YFP and *E. coli* BFP in agarose microbeads. (**a**) 2-channel microscopic images of *E. coli* YFP and BFP growth in agarose microbeads at different time points at inoculum size of 2 cells/bead. Note that the images were taken at a random position. Growth curves of *E. coli* YFP and *E. coli* BFP in monoculture and co-culture (ratio 1:1) conditions were constructed based on (**b**) colony size and (**c**) fluorescence intensity. The insets show the experimental data of *E. coli* BFP and *E. coli* YFP coculture growth and its modeling curve. The fitted curve was obtained by simulating the experimental data based on Equation (1).

**Figure 5 micromachines-14-00645-f005:**
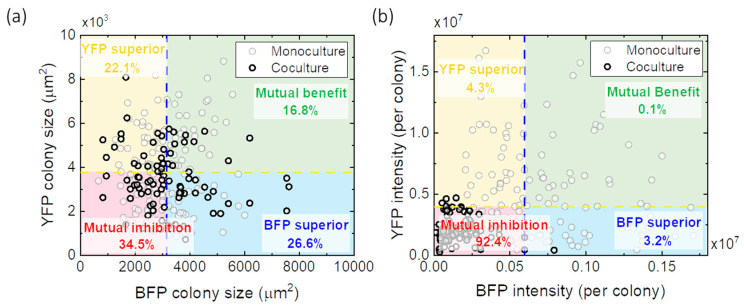
Distribution of interactions between two strains of *E. coli* during co-culture in gel beads based on (**a**) colony size and (**b**) fluorescence intensity. The blue and yellow dashed lines indicate the median value of monoculture results. Grey dots present random pairs of monoculture colonies, and red dots show random pairs of co-culture colonies. The numbers on the figures are the average values of results from 1000 generated random data sets with a standard error < 2.58%.

## Data Availability

Authors confirm that the data supporting the findings of this study are available within the articles and its Appendix A. Raw data that support the findings of this study are available upon reasonable request.

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
