# Peer review of "High-Throughput Gel Microbeads as Incubators for Bacterial Competition Study"

_micromachines, 2023, doi:10.3390/mi14030645_

Round 1

Reviewer 1 Report

The article titled “Gel microbeads as incubators for bacterial competition study” discusses a novel approach to generate micro gel bead reactors for the study of cell interactions at low levels of inocula of co cultures. The authors study colony morphology, effect of inoculum size on colony size, growth kinetics, co-culture interactions within micro gel bead reactors utilizing two types of E. Coli strains that emits yellow and blue, fluorescent signals. The authors claim that their micro gel bead devices are suitable for the study of bacterial interactions of co cultures and as drug screen assays. Overall, the paper suggests a novel approach and provide suitable and potential applications for the developed technology. Hence, the paper is suitable for the publication in the journal ‘Micromachines’.

However, in my opinion minor revision is needed to the content provided to polish up the manuscript and my comments are provided below.

1. Page 3, paragraph 1, line 9: Please provide the surfactants specifications (composition) other than the brand of surfactant.

2. Page 3, paragraph 1, line 10: Please explain the acronyms in their long form as these are used for the first time in the text. It would improve the clarity of the text to the reader.

 3.      Page 3, section 2.2, line 5: Here the E. coli is mixed with agarose at a temperature of 60 ËšC. Does this temperature of 60 ËšC affects the survival of E. coli and inoculum size?

 4.      Page 4, paragraph 2, line 5 (Explanation to Eq. 4, 5, and 6): It would be better to state that subscription B represents BFP while subscription Y represents the YFP strain.

 5.      Page 4, section 3.1, paragraph 1, line 14: The sentence states that the configuration is suitable to study bacterial population dynamics. Why is such a configuration suitable for this purpose? May be connecting to the introduction section where it explained the necessity of this would improve the clarity of this statement.

 6.      Figure 1(c), caption – line 8: Please clarify the histogram in detail. the X-axis stands for colonies/droplet and the text says 2 cells/microbead innoculum. Hence, it would be better to provide a detailed explanation.

 7.      Figure 1(d), caption – line 9: The distribution described here is with respect to which parameter? is it with incubation time? or inoculation ratio?

 8.      Figure S6, caption – line 2: Flow rate is given but has not mention the components. 1:7:3 of which components?

 9.      Figure 2(e), caption -line 7: The colour code used in legend seems to be different from that of histograms.

 10.   Page 8, paragraph 3, line 6 (Explanation to Eq. 12): Please maintain the consistency of the notations used. (Bead and droplet).

 11.   Figure 4(c), caption – line 5: Please revise the description for insets. These insets show data for both BFP and YFP strains.

 12.   Page 11, paragraph 1, line 5, 6, and 7: It would improve the clarity and completeness of the content if authors explain why this difference in fluorescence is observed for mono and cocultures.

 13.   Figure S13 (A) and (B): it would be better if same colours are used in histogram (B) to match with (A). The colour of BFP con seems different in (A) and (B). Even though these two discuss different parameters it would be better to have a connection between them as (A) provides a complete legend to the specimens used where as (B) provides the specimen’s name only partly.

 14.   Figure S15 (A) and (B): Similar to Figure S13. The colours used for BFP+ in (A) and (B) seems different. It would be better if they are consistent for the ease of reader.

Author Response

Reviewer 1:

The article titled “Gel microbeads as incubators for bacterial competition study” discusses a novel approach to generate micro gel bead reactors for the study of cell interactions at low levels of inocula of co cultures. The authors study colony morphology, effect of inoculum size on colony size, growth kinetics, co-culture interactions within micro gel bead reactors utilizing two types of E. Coli strains that emits yellow and blue, fluorescent signals. The authors claim that their micro gel bead devices are suitable for the study of bacterial interactions of co cultures and as drug screen assays. Overall, the paper suggests a novel approach and provide suitable and potential applications for the developed technology. Hence, the paper is suitable for the publication in the journal ‘Micromachines’.

However, in my opinion minor revision is needed to the content provided to polish up the manuscript and my comments are provided below.

We thank the reviewer for this encouraging remark. We address all following remarks/ comments/questions of the reviewer.

  1. Page 3, paragraph 1, line 9: Please provide the surfactants specifications (composition) other than the brand of surfactant.

Thank you for this reminder, we add the composition of the surfactants as 2% (w/w).

  1. Page 3, paragraph 1, line 10: Please explain the acronyms in their long form as these are used for the first time in the text. It would improve the clarity of the text to the reader.

The long form of the acronyms is added as Fluorinated ethylene propylene (FEP)-tubings, Ethylene tetrafluoroethylene (ETFE) T-junction.

  1. Page 3, section 2.2, line 5: Here the E. coli is mixed with agarose at a temperature of 60 ËšC. Does this temperature of 60 ËšC affects the survival of E. coliand inoculum size?

The 60 ËšC was only maintained for less than 1 second during gelation. Once the bacterial medium was mixed with the agarose at a temperature of 60 ËšC, the temperature of the agarose decreased quickly. The frozen bag was also used to reduce the heat quickly. After capsuled the bacteria in the gel beads, the distribution of colony number (Figure. 1c) indicates the initial inoculum size matched the predicated Poisson distribution at certain inoculum. Thus, we believe the short time of high temperature did not affect the bacteria's survival.

  1. Page 4, paragraph 2, line 5 (Explanation to Eq. 4, 5, and 6): It would be better to state that subscription B represents BFP while subscription Y represents the YFP strain.

We add the explanation of B and Y shown in Eq. 4, 5, and 6 as reviewer suggested.

“B represents BFP, while Y represents the YFP strain,”

  1. Page 4, section 3.1, paragraph 1, line 14: The sentence states that the configuration is suitable to study bacterial population dynamics. Why is such a configuration suitable for this purpose? May be connecting to the introduction section where it explained the necessity of this would improve the clarity of this statement.

We correct the sentence to better suit the paragraph as:

“Such configuration is suitable for investigating microbial population dynamics in semi-solid environments in a high-throughput manner”

  1. Figure 1(c), caption – line 8: Please clarify the histogram in detail. the X-axisstands for colonies/droplet and the text says 2 cells/microbead innoculum. Hence, it would be better to provide a detailed explanation.

We explained more details of the histogram in the main text as: “Histogram of bacterial colony number distribution per microbead (droplet) in experimental data and predicted Poisson distribution, showing the number of bacterial colonies per microbead matched the predicated Poisson distribution at an inoculum of 2 cells/microbead”.

  1. Figure 1(d), caption – line 9: The distribution described here is with respect to which parameter? is it with incubation time? or inoculation ratio?

We correct and add more details:

(d) Distribution of E. coli YFP and E. coli BFP colonies in microbeads at an inoculation ratio of 1:1 and an inoculum size of 2 cells/microbead. The color scale from blue to red represents the probability of having a specific colony number of BFP (0-6) or YFP (0-6) cases in the droplet, e.g., the statistical probability of having one BFP and one YFP is 14.1%.

  1. Figure S6, caption – line 2: Flow rate is given but has not mention the components. 1:7:3 of which components?

We added details for the components:

“(b) microbead size at the chosen flow rates 1:7:3 (HFE oil: mineral oil: aqueous phase)”

  1. Figure 2(e), caption -line 7: The color code used in legend seems to be different from that of histograms.

We ask the reviewer for understanding. We used the Origin software to prepare the graph thus the legend is auto-generated and should be matched to the color of the graph. In this case, we used transparent color to make overlapping possible thus the color is a little bit off, especially where overlapping occurred. However, we have checked this and believe that it is not affecting the readability of the graph.

  1. Page 8, paragraph 3, line 6 (Explanation to Eq. 12): Please maintain the consistency of the notations used. (Bead and droplet).

In this paper, we refer to our system as “gel bead” or “microbead”. In this paragraph, we compared the number of generations to a different system of “liquid droplets” published in the reference thus the word “droplet” refers to the other system.

“The numbers correspond to those in liquid droplets with similar glucose concentration [24].”

  1. Figure 4(c), caption – line 5: Please revise the description for insets. These insets show data for both BFP and YFP strains.

We revised the decription for inset as:

“The insets show the experimental data of E. coli BFP and E. coli YFP coculture growth and its modeling curve”

  1. Page 11, paragraph 1, line 5, 6, and 7: It would improve the clarity and completeness of the content if authors explain why this difference in fluorescence is observed for mono and cocultures.

We add a brief explanation to the sentence.

“We saw that both strains had higher fluorescent intensity in monoculture than in co-culture which suggests mutual inhibition interaction in co-culture condition.”

  1. Figure S13 (A) and (B): it would be better if same colours are used in histogram (B) to match with (A). The colour of BFP con seems different in (A) and (B). Even though these two discuss different parameters it would be better to have a connection between them as (A) provides a complete legend to the specimens used where as (B) provides the specimen’s name only partly.

We correct the graph to match the color as suggested by the reviewer.

  1. Figure S15 (A) and (B): Similar to Figure S13. The colours used for BFP+ in (A) and (B) seems different. It would be better if they are consistent for the ease of reader.

We correct the graph to match the color as suggested by the reviewer.

Reviewer 2 Report

The manuscript reports a new type of nanoliter-volume micro gel beads that is positioned as a platform to study competitive and cooperative interactions between two strains of the bacterium Escherichia coli, growing as colonies. The contents are interesting and fall well within the scope of the journal. I recommend its acceptance for publication after minor revision.

A separate CONCLUSIONS section is needed.

The figures in the insets of Figure 4b and 4c are too small, please enlarge them for an easy reading.

The references formats please be unified according to the journals request.

Author Response

Reviewer 2:

The manuscript reports a new type of nanoliter-volume micro gel beads that is positioned as a platform to study competitive and cooperative interactions between two strains of the bacterium Escherichia coli, growing as colonies. The contents are interesting and fall well within the scope of the journal. I recommend its acceptance for publication after minor revision.

A separate CONCLUSIONS section is needed.

Since most of the discussions are integrated into section “3. Results and Discussion”, we would like to rename the last section as “4. Conclusion”. This section provides a summary of the works and our perspective on the outlook for the system.

The figures in the insets of Figure 4b and 4c are too small, please enlarge them for an easy reading.

We agree with the reviewer about the readability of the inset. We enlarged the inset as suggested by the reviewer.

The references’ formats please be unified according to the journal’s request.

We fixed the reference list according the format of the journal.

Reviewer 3 Report

In this study, the author presented a technique to produce numerous agarose microbeads that replicate the natural environmental conditions for bacteria. These beads are used to co-encapsulate two varieties of fluorescence-labeled Escherichia coli. The ultimate size of the colonies is negatively correlated with the inoculum size and the shape and fluorescence intensity of the colonies in monoculture and co-culture vary significantly. A model was compared with the experimental results. In my opinion, this small-scale solid bioreactor could prove to be useful in investigating the characteristics of interactions among bacterial populations with high precision in an environment that mimics natural spatial conditions to some extent.

The manuscript was well-written and organized. I just suggest the author move the letters that mark the order of the graphs to the top left position of each graph in Fig. 1 and Fig. 3. Also do it with the supporting file, use the same font, and same lowercase for consistency.

It is suitable to be published in its current form after editing those very minor points!

Author Response

In this study, the author presented a technique to produce numerous agarose microbeads that replicate the natural environmental conditions for bacteria. These beads are used to co-encapsulate two varieties of fluorescence-labeled Escherichia coli. The ultimate size of the colonies is negatively correlated with the inoculum size and the shape and fluorescence intensity of the colonies in monoculture and co-culture vary significantly. A model was compared with the experimental results. In my opinion, this small-scale solid bioreactor could prove to be useful in investigating the characteristics of interactions among bacterial populations with high precision in an environment that mimics natural spatial conditions to some extent.

The manuscript was well-written and organized. I just suggest the author move the letters that mark the order of the graphs to the top left position of each graph in Fig. 1 and Fig. 3. Also do it with the supporting file, use the same font, and same lowercase for consistency.

We fixed the letters that mark the order of the graphs in the main text as well as the supporting information as suggested by the reviewer.